# Responses to Apocalypse: Early Christianity and Extinction Rebellion

## Cullan Joyce

Department of Philosophy, Catholic Theological College, University of Divinity, East Melbourne, VIC 3002, Australia; cullan.joyce@ctc.edu.au

**Abstract:** The Extinction Rebellion (XR) movement has grown rapidly in the past two years. In popular media, XR has sometimes been described using religious terminology. XR has been compared to an eco-cult, a spiritual and cultural movement, and described as holding apocalyptic views. Despite XR lacking the distinctive religiosity of new testament and early (pre-150ACE) Christianity, the movement resonates with the early Christian experience in several ways. (1) A characterization of events within the world as apocalyptic. (2) Both feel vulnerable to the apocalypse in specific ways, though each responds differently. (3) Both experience the apocalypse as a community and develop community strategies in response to the apocalypse. The paper sketches certain features of new testament Christianity and compares some of these to XR. The main difference between the two movements is that XR makes decisions to actively become vulnerable, whereas new testament Christianity was more often passively vulnerable. Elements of new testament Christianity provide a context for understanding XR as a response to an apocalypse.

**Keywords:** climate crisis; apocalypse; extinction rebellion; Christianity; active love; hope; civil disobedience

## 1. Introduction

*The Point Is Not to Change the World, But to Save It*

The Extinction Rebellion (XR) was formed by a group of activists in the UK in October, 2018 (Extinction Rebellion 2020).[1] The movement has expanded to 69 countries, and although numbers are difficult to quantify, XR's Action Network has over 30,000 listed members in Australia and at least 9000 in the state of Victoria.[2] XR is not a religious movement. However, some elements of the movement have led to it being described using terms connected to religious traditions (Campbell 2019; Skrimshire 2019; Charles 2020). Connections between the Extinction Rebellion (XR) and Christianity have been mulled over by commentators (Pihkala 2019). Christians have been part of, and have participated in, XR (Zhang 2019). One explanation for the use of religious terminology when characterizing XR could be XR's unabashed apocalypticism; (Collins 2014)[3] it experiences climate change (Intergovernmental Panel on Climate Change  IPCC)[4] and mass extinction (Ceballos et al. 2017; Grens 2017) together

---

1. https://rebellion.global/about-us/ (accessed on 15 July 2020).
2. https://actionnetwork.org/ (accessed on 15 July 2020). This data is not publicly It was supplied to me by one of the founders of XR in Australia, Jane Morton (Conversation with Jane Morton, 14 July 2020).
3. "Perhaps the most obvious way in which apocalypticism persists in the modern world is the recurring expectation of an imminent end of history or of the world itself." (Collins 2014, p. 9).
4. The Intergovernmental Panel on Climate Change (IPCC) reported in October, 2018 that we must reduce carbon emissions by 40% in the next 12 years to have a 50% chance of avoiding 'catastrophe'. Climate change is causing wide-spread effects across a variety of areas, including food security, the health of oceans and fresh water sources, migration, and markets.

as an apocalyptic event that cannot be evaded, fixed by adopting new world views (Lent 2016) or left for future generations to fix. Instead, climate change and mass extinction make up a scenario that calls upon individuals to become fully awake (Monbiot 2019) to facts of impending catastrophe, while demanding that governments pass legislation that will bring about "deep decarbonization" (Gunningham 2019) as soon as feasible, and halt biodiversity—animal, plant, and insect—loss[5]. Roger Hallam, one of XR's co-founders, writes that "Extinction Rebellion was set up in April 2018 to tell the truth and act as if that truth is real" (Hallam 2019). In other words, digesting and broadcasting scientific predictions of damage to living organisms on an awesome scale should be accompanied by thoroughly restyled behavior.

XR advocates nonviolent, mass civil disobedience as one way of triggering swift and society-wide restructuring. Disobedience tactics usually involve people placing themselves in disruptive and vulnerable situations—for example, by sitting in roads and blocking traffic. Jem Bendell, Professor of Sustainability Leadership at the University of Cumbria (UK), provides a summary of why XR takes this approach, referring to urgency of action and the failure of slow, progressive reforms to generate the rate of change required to head off the effects of climate change[6]. It could be put like this: XR desires, in T. S. Eliot's words, "to save the World from suicide" (Eliot 2015).

## 2. Tell the Truth

### 2.1. How Did Early Christians Understand the Apocalypse?

This section discusses some ways that new testament Christianity (37–100ACE) experienced cataclysms—literally, a deluge or destructive event—as apocalyptic: as an event that reveals something. First, it examines the relationship between Apocalyptic and Eschatological Christianity before it discusses how Christian watchfulness, or the close observation of cataclysms, provides the catalyst for revealing the truths of God's kingdom. It argues that the church is often a vulnerable observer of events outside of its control and discusses watchfulness considering this. The discussion of the tone of vulnerable watchfulness of new testament Christianity (hereafter, NT Christianity) will be contrasted with how XR experiences ecological breakdown as demanding active vulnerability.

"Vulnerability" is used in a personal and community context. The term refers to how a person or a community, due either to their disposition or context, finds themselves in a position where their capacity for autonomous action and self-determination is reduced (Rogers et al. 2012). This means that the person or community's well-being (in a broad sense) becomes more reliant on factors outside of their control; they are not "inherently" vulnerable, but become so when compared to the capacity of another citizen or community, unaffected by the circumstances that define the vulnerable individual/community. Therefore, being vulnerable is not identical with experiencing suffering (broadly defined); instead, vulnerability refers to a person or community's situation or disposition, as we shall see below. Moreover, as mentioned, sometimes circumstances or dispositions are entered into intentionally. At other times, these circumstances are not chosen and, therefore, the individual or community finds itself being *made* vulnerable.

Why use the term "apocalypse" rather than "eschaton" and eschatological language to describe Christianity? An apocalyptic religion is one that experiences the present as revealing something of God's plan, often mediated through the voice of a prophet[7]. An eschatological faith sees the divine plan becoming evident (A) as articulated in (Christian) scripture (in the return of Christ), and (B) as

---

Unless drastic changes are made, it is unlikely that the rise in global average temperatures could be limited to 1.5 C. *Global Warming of 1.5 C*. Intergovernmental Panel on Climate Change (IPCC). Available online: https://www.ipcc.ch/sr15/ (accessed on 10 March 2020).

[5]　https://rebellion.earth/the-truth/about-us/.

[6]　https://www.youtube.com/watch?v=Ke1PCgiM3V4 (accessed on 27 May 2020).

[7]　(Rowland 1996, pp. 238–40, 242).

the realization of the completion of the cycle of salvation history (Menn et al. 2018)[8]. New testament Christianity is, in part, an interconnection between the apocalyptic and eschatological elements (Dunn 2006, p. 337). Jesus experiences events of the times as heralding some culmination of God's plan, although it is debatable as to whether he thought there was a complete culmination occurring, mediated by the "Son of Man" (which can be understood as referring to Jesus Christ). Rausch describes Christianity as eschatological, noting how the Christian faith experiences the sacramental present in an eschatological light (Rausch 2012). As such, the apocalypse evokes the eschaton, especially as it references the possibility or the realities of the end of history (at least for the believer). For Christianity, specifically in the resurrection, the "now" reveals the end of history destined or brought about—in this case, in the person of Christ (Bryan 2011, p. 4). Persecutions were also seen to connect the eschaton with the apocalyptic reading of historical events: "the righteous dead would be restored to life in order to share the blessings of the new age . . . " (Dunn 2006, p. 342). In the present, historical events are eschatological insofar as they show how vulnerable history is to reshaping (Kelly 2006, pp. 21–22). The eschatological or "end times" discourse is concerned with the end of history, and, certainly, the climate crisis has a sense of the end times, insofar as people conceive that the collapse of ecosystems might prevent human histories from occurring. However, our decision to use the apocalypse rather than eschaton reflects the sense that the apocalyptic experience of history is an experience that is less defined and less specific than the eschatological one. Eschatology, in a Christian setting, reads history as a Christ-centered culmination, whereas an apocalyptic experience of present events sees events as revealing something of God's plan, but not necessarily a culmination of a specifically Christ-centered experience (Collins 2016).

The new testament frequently represents events as a manifold: cataclysmic events become apocalyptic when they express a significance to the salvation history initiated by/in Jesus. In support of this, discipleship in Mark is a mixture of an apocalyptical outlook and the practical example of Jesus (Vena and Myers 2014, chp. 1). In the face of new ways of being, embodied by Jesus, Jesus represents a radical challenge to many of those who encountered him[9]. For example, when the rich young man does not follow Jesus, we are given a glimpse into the man's internal states: "At this the man's face fell. He went away sad, because he had great wealth". Jesus is a catalyst for the revealing of the disciple's internal states: "He [Jesus] said to his disciples, 'Why are you so afraid? Do you still have no faith?' They were terrified and asked each other, 'Who is this? Even the wind and the waves obey him!'" (New International Version)[10]. The painfully vulnerable internal states of Jesus' disciples and others become revealed in the context of their response to the challenge of Jesus.

Chapter 13 of Mark (Bible 2011) depicts Jesus as the agent who can reveal the true meaning of cataclysmic events. The chapter describes the destruction of the temple and Jerusalem, people fleeing to the hills, and the overturning of the natural order of things. The character of these events may not be special: "The Christian gospel came first in an age of terror, when multitudes and, particularly, the authorities were vexed to nightmare by the world in which they lived and by a particular rocking cradle [the birth of Jesus]" (Forrester 2005, p. 11). The new testament church was surrounded by events they could rarely resist. We return to this point below. Suffice to say that the church experiences a forced vulnerability, or victimhood, rather than a vulnerability that is intentionally entered into.

Events do not necessarily convey anything about a divine plan; however, under certain conditions and Jesus' interpretation, events change. Mark 13 frames the cataclysms and changes, including environmental events, around a specific event: the coming of the Son of Man. Jesus is a catalyst for the revealing of experience. The biblical text shows a concrete event as having traces of other significance; the implication is that such catastrophes, though significant, are also signs of something

---

[8]　The second coming is a definite event, prophesied in scripture, which expresses itself in Christ's return.
[9]　Mark 10: 17–22. All biblical translations are New International Version: https://www.biblegateway.com/ (accessed 6 May 2020).
[10]　Mark 4: 38–41.

else (Matt 13: 3). Jesus provides an interpretive lens that describes the overturning of the normal continuity of time (Stein 2014, pp. 128–31) and the emergence of something new: "Now learn this lesson from the fig tree: As soon as its twigs get tender and its leaves come out, you know that summer is near. Even so, when you see these things happening, you know that it is near, right at the door".[11]. Events become parables and reveal new meanings. Through Jesus, events that were purely destructive herald the coming of the Son of Man: events become a manifold (Luke 10: 21–22). Thus, under the sight of Jesus, events change from being only cataclysms (Mark 13: 7–9) to having an apocalyptic (revealing) role.

*2.2. From Cataclysm to Apocalypse*

The sources in the introduction describe the grim situation that climate change places us in, presenting a picture of cataclysmic change—already occurring—that threatens the stability of the earth's ecologies. Once again, for XR, the experience of a climate and ecological apocalypse requires actively intervening through acts of nonviolent, mass civil disobedience (hereafter, simply, civil disobedience). These actions and the movement itself are underpinned by regenerative values[12]. XR uses distinctive apocalyptic language that conveys that the cataclysms are arising due to climate change, and biodiversity losses are profound and have meaning for every human being alive today. In XR's view, one problem that obstructs people from acting is a denial of how desperate the situation actually is. To grasp the import of climate change and mass extinction is to confront a type of death that seemingly defies limits—spatial, temporal, and geographical. Personal mortality is swamped by the concept of the mortality of life itself. Because of the scale of ecological breakdown, in that it implicates and impacts almost every person on the planet today, XR recommends a "universalist approach", which Hallam describes as a "struggle . . . to save all human beings and fulfil our transcultural duty to create a world safe for our children" (Hallam 2019, p. 8).

Accepting our situation requires accepting vulnerability and being willing to experience the grief that can arise when the fuller implications of climate change enter one's experience. The first vulnerability is epistemic, whereby a person is confronted by the ecological situation and decides to take personal responsibility and act. "Knowing" is not a binary concept when it comes to grief-inducing catastrophic information. Indeed, climate-related grief is a condition that some argue follows the well-known "stages" of normal grief and has been observed in a variety of communities, though individual responses to climate research vary (Cunsolo and Ellis 2018, pp. 275–81). XR has mental health resources[13] and there are several different ways that individuals and groups cope and assist with climate grief, including so-called "grief circles"[14]. XR presents material on climate change

---

[11]　Mark 13: 28–29.
[12]　XR's principles are:

1. We have a shared vision of change: Creating a world that is fit for the next seven generations to live in.
2. We set our mission on what is necessary: Mobilising 3.5% of the population to achieve system change—such as 'momentum-driven organising' to achieve this.
3. We need a regenerative culture: Creating a culture which is healthy, resilient and adaptable.
4. We openly challenge ourselves and this toxic system: Leaving our comfort zones to take action for change.
5. We value reflecting and learning: Following a cycle of action, reflection, learning, and planning for more action. Learning from other movements and contexts as well as our own experiences.
6. We welcome everyone and every part of everyone: Working actively to create safer and more accessible spaces.
7. We actively mitigate for power: Breaking down hierarchies of power for more equitable participation.
8. We avoid blaming and shaming: We live in a toxic system, but no one individual is to blame.
9. We are a nonviolent network: Using nonviolent strategy and tactics as the most effective way to bring about change.
10. We are based on autonomy and decentralization: We collectively create the structures we need to challenge power.

[13]　https://rebellion.earth/act-now/resources/wellbeing/ See the Don't Panic booklet. Accessed on 18 May 2020.
[14]　https://rebellion.earth/event/grief-circle-2/ (accessed on 18 May 2020) (Similar resources: https://christianclimateaction.org/resources/climate-grief-and-anxiety/ accessed on 18 May 2020).

through introductory talks[15]. These talks are often rather confronting and, for some, even destabilizing. After absorbing "the truth", XR recommends that people join its nonviolent endeavor to safeguard life, recognizing that no one individual is to blame (see XR's eighth principle in the footnotes). XR does not advocate civil disobedience as a solution to the experience of climate grief, but, anecdotally, civil disobedience does help many participants move through their "eco-anxiety" (Ojala 2018, pp. 10–15) and heartache. Furthermore, accepting the truth and acting accordingly will usually prompt dramatic—not cosmetic—changes in a person's life. For example, acting in accordance with an apocalyptic framework may endanger relationships with family, friends, and co-workers. It may lead to a change of job, career, or forsaking "career games" altogether (Bendell 2018)[16]. Responding to the ecological apocalypse, then, XR's message is "existential" in the following ways: (1) climate change is a threat to the existence of life on earth, and (2) the decision to confront and take responsibility for climate change and mass extinction is of existentialist magnitude for an individual.

*2.3. Summary*

Both NT Christianity and XR have an openness to seeing cataclysm as the bearer of great significance. Both movements are forms of response to the apocalypse. Such responses give rise to public forms of vulnerability, though the causes of the vulnerability differ. Both groups, therefore, emphasize the need to face and play an active role in upheaval.

## 3. Act as If the Truth Is Real

*3.1. How Did Christianity Respond to Immanent Threats in the Form of Persecutions?*

The vulnerability in martyrdom is not a general vulnerability to the world or God; it is a response to persecution from an external agent, such as the state [17], through various means, including legally sanctioned persecution (De Ste-Croix 2006, pp. 109–11). Christian communities often had a sense of themselves as suffering or persecuted, although the meaning of "persecution" can be quite broad. Some articulations were not just responses to external persecutions, but attitudes and self-characterizations made by the communities. The Christians associated themselves with the life and suffering of Jesus (Moss 2012, p. 51). Martyrdom is understood in different ways and can be applied to different circumstances (Moss 2012, pp. 1–3) and changes between Jewish uses of the term (or equivalent) and new testament Christian uses (Grayston 1996, pp. 250–51; Goehring 1997, p. 129)[18]. Martyrdom as we are using it was a response to external circumstances, and had great, heroic (Gemeinhardt and Leemans 2012, pp. 9–10), liturgical (Young 2001, pp. 9, 21), sacramental, and soteriological significance[19].

In popular use, a martyr is a person who witnesses to religious faith under the threat or promise of death. Martyr does not always refer to a circumstance where a Christian provides a witness to their belief in Jesus in the face of their immanent death. The term martyr can include those who witness to their faith and confess their belief (in Jesus) in a public way. Martyrdom is an act that is modeled on the sacrifice made by Jesus (Eastman 2015, pp. 2, 15). Ferguson traces the origins of the Christian use of the term to the synoptic and Johannine Gospels, where related terms are used to designate both to witness the resurrected Christ, and the testimony on behalf of Christ (Ferguson 1997). This paper

---

[15] (https://www.facebook.com/events/zoom-event/heading-for-extinction-and-what-to-do-about-it/1618398331657721/ accessed on 18 May 2020).

[16] As Jem Bendell puts it: "We no longer have time for the career games of aiming to publish in top-ranked journals to impress our linemanagers or improve our CV for if we enter the job market. Nor do we have a need for the narrow specialisms that are required to publish in such journals." "Deep Adaptation: A Map for Navigating Climate Tragedy," IFLAS Occasional Paper 2, www.iflas.info (July 2018): pp. 1–36, 24. http://www.lifeworth.com/deepadaptation.pdf. accessed on 9 June 2020.

[17] (Cunningham 1997, pp. 257–58).

[18] (Grayston 1996, pp. 250–51; Goehring 1997, p. 129).

[19] Ferguson, *Baptism.* and Ferguson, *Confessor*, in: *Encyclopedia of Early Christianity*, 164 and 274.

focuses on two variations of "martyrdom" as an act of (a) public witness, which is (b) made under a threat of, or evident, persecution.

Ferguson follows authors, such as Tertullian, who refused to describe martyrdom as a form of political resistance. Although there are instances where persons voluntarily accept or undertake martyrdom (De Ste-Croix 2006, pp. 153–55, 162–63), Ferguson does not think that Christians were engaged with forms of nonviolent resistance or civil disobedience Catechesis, baptism, eschatology, or martyrdom (Ferguson 2014, pp. 269–70, 278–79). Though Roman authorities considered witnessing a form of civil disobedience, Ferguson argues that martyrdom is a witness that leads to persecution and death. Hence, martyrdom is first a witnessing to, or a confession of, faith in the resurrected Christ, and is only accidentally a form of civil disobedience. New testament Christian communities experienced persecution less as a catalyst for spiritual combat or an opportunity for testing the moral strength of the martyr, or even the defense of the church as such. Instead, witness is a representation of the Christian's receptivity and vulnerability to the unfolding of salvation history. Martyrs are particularly vulnerable not just to authorities, but also to spiritual entities and temptations (Young 2001, pp. 9–10).

Let's take a concrete example: in the lead up to Jesus' death and resurrection, as it is described in the Gospel of Mark, Jesus is in Jerusalem and has been preaching. He is aware of the danger that lies before him in exact terms—trial, crucifixion, death—and has said that this will culminate in his glory. The narrative comprises, then, the impending event, Jesus' sense of it, and then his responses during his final days in Jerusalem. One response was to describe future persecutions (Mark 13). After discussing future disasters and persecutions, Jesus describes how one should relate to these:

> "But about that day or hour no one knows, not even the angels in heaven, nor the Son, but only the Father. Be on guard! Be alert! You do not know when that time will come. It's like a man going away: He leaves his house . . . and tells the one at the door to keep watch. Therefore, keep watch because you do not know when the owner of the house will come back . . . If he comes suddenly, do not let him find you sleeping. What I say to you, I say to everyone: 'Watch!'" (Mark 13: 32–37)

There is a relationship between vulnerability and watchfulness. The Greek γρηγορεῖτε can be translated as be alert, stay awake, keep watch. The word is also used in Matthew 26: 41: "Watch and pray [ . . . ]" (γρηγορεῖτε καὶ προσεύχεσθε). Ὑρηγορεῖτε, in this context, has a sense of looking out for something to see what might occur. The believer is attentive to the unfolding of events (including persecutions) so that they may discern whether such events are connected to Jesus' words and salvation history. However, watchfulness is not limited to discerning persecutions.

In a different context, the term θεάομαι means to look or "see", and, with derivatives, to behold. The synoptic Gospels use the term in the context of witnessing the resurrection; John uses it when the narrator beholds the logos. Other terms, such as some derived from θεωρέω, are also used to denote seeing with the connotation of comprehension. Moreover, γρηγορεῖτε as a disposition that the Christian has is connected to a range of terms, including witnessing or seeing Jesus. In this sense, the response of the believer to watch does not only mean "wait for calamity", but includes a sense of paying attention to, or watching for, Jesus. The sense of "watchfulness" might have a sense of anxiety, but also prayerful and reflective connotations. Overall, the various connected terms imply that watchfulness can be attitudinal. This attitudinal aspect arises because there is an extended range of phenomena at play (Jesus, God). Watchfulness is part of a broader context of habituations and awareness and includes openness to God in various guises. In the context of the apocalypse and persecution, watchfulness is fundamental for the Christian believer as they attempt to experience Jesus in unfolding events.

### 3.2. The Apocalypse and Community Fellowship

Peter's Speech (Acts 2: 14–40) uses the scripture beginning "In the last days . . . ": the scripture describes the effects on the people who will prophesy and have dreams, and there will also be wonders (τέρατα) and signs (σημεῖα) that reshape the world. In this context, the foremost event is the death and resurrection of Christ, who, raised, now sits at the right hand of God. The people ask Peter: what shall we do? He replies, "repent and be baptized". Μετανοήσατε refers to a change of mind or a reorientation of knowing that baptism is for repentance, the forgiveness of sins, and is an initiation into the community of believers. This indicates how repentance and entry into the community both make radical demands on the new believer.

Peter preached that the end of history is immanent: in other words, he sees that the community he is speaking to is nearing the apocalypse. The call to listen to the teachings, repent, sacrifice personal property, and maintain communion with one another in the church's liturgical expressions in the temple and in people's homes. Communion, as both a sharing of property and liturgical communion, is a response to the revelation of the end of history. Following Acts 2: 42–47, repentance and entry into the community had several responsibilities: dedicated perseverance (προσκαρτεροῦντες) in maintaining the apostles' teaching and fellowship (κοινωνίᾳ). Adding to these, holding property in common (κοινά) and selling possessions were expectations of fellowship (κοινωνίᾳ). Ferguson describes how κοινωνίᾳ fellowship flows across many of the activities of the early church (Ferguson 2008). Fellowship—community—involves attitudinal changes and practical sacrifices. This is a deep sense of the communal. Furthermore, Acts 5 shows the results of individualism when the husband and wife choose not to give the Christian community all of their possessions. The text further describes how the community remained dedicated in their common passion προσκαρτεροῦντες ὁμοθυμαδὸν by gathering in the temple and in each other's homes to break bread. The term προσκαρτεροῦντες is translated as to be devoted to, to attend to constantly, and has a sense of perseverance in that connection. The term does not have the exact sense of "remain", or persevere, but the connotation is that the consistent devotion to the breaking of bread, prayer, and liturgical gatherings is similar to the perseverance in the face of persecution, of which Jesus and the Apostles were exemplars (Avemarie and van Henten 2002, p. 89). Sharing in prayer and common meals would build the connection of the community so that perseverance in persecution was built by these communal acts. The believers undertook to maintaining property in common and the believers sold their possessions to give to the poor. "All the believers were one in heart and mind. No one claimed that any of their possessions was their own, but they shared everything they had" Acts 4: 32[20]. The meaning of "common" κοινά can denote a collective experience of liturgy, collective or community resilience, and holding property. The breaking of bread is both an act of sharing in common and a reenactment of the last supper: the meal before Christ's death.

Ferguson notes distinctive elements of the Eucharist in the new testament (Ferguson 2008, chp. 7). Elsewhere, he discusses the frequency and importance of the phrase "when you come together" (ἐπὶ τὸ αὐτὸ Acts 2: 44) in scripture and early Christian sources (Ferguson 2014, p. 17). The phrase was interpreted in many ways: as a description of literal proximity, as a model for believers, and as an image for the church itself. The majority of uses seem to be quite literal in describing how believers are to meet together. Hence, being together is crucial context. In the context of Luke Acts, the meal brings people together with an eschatological emphasis. Heil discusses how the Lukan sources (Luke and Acts) drew the connection between the meals Jesus shared with disciples and the community (including "sinners") and Isaiah's eschatological banquet (Heil 2018, chp. 7). Sharing the literal meal with sinners, the poor, and the just alike offers a glimpse of the eschatological banquet. Jesus feeding the crowds and the wedding at Cana shows that the meal is where Jesus displays who he is. If the

---

[20]  Τοῦ δὲ πλήθους τῶν πιστευσάντων ἦν καρδία καὶ ψυχὴ μία, καὶ οὐδὲ εἷς τι τῶν ὑπαρχόντων αὐτῷ ἔλεγεν ἴδιον εἶναι, ἀλλ' ἦν αὐτοῖς πάντα κοινά.

apocalypse and persecutions are events that the community must endure passively, then holding common property, turning up for prayer, and liturgy are active forms of communion.

### 3.3. Persecution and Endurance

As mentioned in the previous sections, watchfulness is part of seeking out the signs of the Son of Man. We described how, in the light of Jesus, cataclysms become apocalypses (manifolds or events, in phenomenological terms). The experience of suffering does not necessarily have any redemptive or meaningful connotations, so the complexification of the experience of cataclysm and persecution is an outcome of at least two factors: what Jesus said and who he is and the historical situation of the Marcan community. The Marcan community would have experienced significant changes (warfare, economic, or religious persecution) as outside of their control. Both Jesus and the earliest Church experience events as persons living within the hand of Roman colonial power (and terror) (Leander 2013).

Whilst the apocalypse was a threat, it was something that could also be observed from a distance (though not always), but persecution is different: it is direct. Persecution inflicts pain on the communities and causes that community to be vulnerable, as the community's physical, emotional, social, and religious space come under direct attack. In this context, Mark 13 describes how the believers need to endure (ὑπομείνας εἰς τέλος οὗτος σωθήσεται) until the end will be saved. The verb, ὑπομείνας, does not appear frequently in the NT; when it does, it is applied almost identically. Matthew 10: 22 to Jesus describes the need to endure in the face of persecution: "You will be hated by everyone because of me, but the one who stands firm to the end will be saved"[21]. The term more commonly used in the NT, ὑπομένω, refers to remaining in the sense of staying put and enduring, and is used in some Pauline letters as enduring persecution[22]. The term in Mark is concerned with endurance in the face of persecution. The question is, what is the tone of endurance: is it a form of resistance? Is it passive? How is it shaped by the apocalyptic context? Mark places these things together: apocalypse, watchfulness (for signs of the Son of Man), and the persecution of the community. Endurance is part of the response that the community has to what is occurring to them[23]. If watching has a connotation of detached observation, enduring is a way of responding to something that affects your community and body. Both apocalypse and persecution contain signs of an emerging event; as such, endurance, like watchfulness, is a way of relating to these events and experiencing them not as meaningless suffering, but as containing something else. Endurance, therefore, is a purposive resilience, rather than just putting up with something.

### 3.4. Summary

Christianity is not a passive religion, though it emerged in a context where it seemed improbable that it could affect the Roman world; it experienced cataclysms as apocalyptic and non-Christians actively decided to join the church. Members of the church willingly gave their property for common use and entered the restrictions of common life. For NT Christianity, entry into the community requires giving away possessions and having meals together. This means that the new community member must accept a loss of power and increased vulnerability. Both Christianity and XR experience vulnerability in different ways. As discussed below, XR experiences different forms of vulnerability: some actively enter and some are passive. We note some of the vulnerabilities that define XR and discuss how resilience helps individuals and communities to maintain long-term vulnerability that comes with civil disobedience. For XR, the community experiences vulnerability to climate change through the activity of civil disobedience, and through regenerative engagement with community.

---

[21] https://biblehub.com/greek/strongs_5278.htm (accessed on 6 May 2020).
[22] https://biblehub.com/greek/strongs_5278.htm (accessed on 6 May 2020).
[23] *The Didache* XVI, 5. The text echoes this pattern (https://www.ccel.org/l/lake/fathers/didache.htm, accessed on 6 May 2020).

## 4. XR's Three Types of Vulnerability

Mass rebellion is a response to a comprehension of cataclysmic climate change and mass extinction. To remedy climate change and biodiversity loss nonviolently requires accepting vulnerability and weakness in the face of entrenched societal habits (such as driving), powerful institutions, indifference, and so on. This vulnerability at the heart of the movement comes from the acceptance of the reality of the apocalyptic magnitude of climate and ecological deterioration. The movement's choice of civil disobedience makes acceptance of one's individual powerlessness. Our reconnection to each other opens news depths and challenges us to be better. If one part is missing—the scientific truth, civil disobedience, community—the movement loses something of its core. Sacredness is an emergent property arising from these different vulnerabilities. Sacredness is present in the sense of the significance of the climate apocalypse and our response to these circumstances, and sacredness infuses the acts of civil disobedience and other moments of vulnerability that constitute the movement (Eisenstein 2013). Sacredness here is like an energy or creative order (Streng 2019): transcendental to these experiences of vulnerability, but without being something that members of XR deliberately seek to replicate (we are not necessarily religious in our sensibilities).

### 4.1. Vulnerability to the Truth of Climate Change

Previously, we discussed vulnerability to cataclysm as a vulnerability to truth. In this section, we will examine how XR's experience of cataclysm becomes akin to an experience of apocalypse along the lines of the Christian understanding. We argue that, for XR, catastrophic climate change is an experience of an apocalypse, not just a cataclysm. The basis for this is the change of behavior that occurs as a result of someone being vulnerable to the truth. In this way, although the language of "watchfulness" is absent, the qualitative change that XR asks of people plus the change in personal behavior that can and does often occur signal that we are in the realm of apocalypticism.

The evidence that XR's message is apocalyptic—which is to say, revealing—is that it has initiated changes in the outlooks and behaviors of many members. The evidence for the revelatory character of XR's explanation of the climate crisis is that it provides the basis for civil disobedience. People join XR and then they do things that they never believed themselves capable of before, such as breaking the law. There are two reasons for this, which we list in order of importance: (1) the evidence of climate and ecological cataclysm; (2) XR's suggested pathway into active response. First, climate change and the problem of dizzying biodiversity loss are real. Second, XR's presentation articulates a response to the cataclysm. Civil disobedience is a response to the truth of the climate crisis. Some XR members describe their responsibility as being a duty; as something they feel impelled to do[24]. XR emphasizes taking an active pose in relation to the apocalypse.

### 4.2. Civil Disobedience as Intentional Vulnerability

What does vulnerability mean within the context of civil disobedience? Is it equivalent to a form of witness or martyrdom? For XR, the civil resistance model—mass-participation civil disobedience—involves communities of activists intentionally making themselves vulnerable in order to disrupt "business as usual". What does this usually look like? Thousands of people peacefully blocking the centers of cities to demand nation-wide change. For XR, the climate apocalypse is a challenge that cannot be addressed by individuals alone, but only in community. Unlike NT Christians, XR's civil disobedience is not a receptive witness to the apocalypse or the end of history. It is a rebellion against the possible end of human history due to climate change. XR members' willingness to be disruptively vulnerable may result in hunger strikes, public vigils, or civil disobedience. The willingness to expose one's vulnerability in a fearless way is the key driver of an emotional

---

[24]　(Conversation with XR Member, Andy G, 23 November 2019.)

response from observers. Hallam has argued that XR's "disruption" must be combined with our willingness to show our vulnerability and to suffer. Protesters who offer to be arrested as part of an action are placing themselves in a space of deep—even spiritual—vulnerability and many are unprepared for the vulnerability that comes with being arrested. To choose nonviolence involves trusting in a society that is often violent toward vulnerable people and ecosystems (MALS Melbourne Activist Legal Support). The disruption, then, simply sets the stage for the symbolism of fearless sacrifice. It is the sacrifice that brings about the social change, not the disruption in itself (Hallam 2019, p. 39).

What is interesting about this view is that it sees disruption as merely a stage for sacrifice, and it is the sacrifice that makes an impact. A person places themselves on an intersection and waits for arrest; someone locks themselves onto a concrete block, or a pole; someone glues themselves to the road, or to the window of a mining company. Authorities can, if they wish, harm the protester(s)[25].

"Mass action cannot just be nonviolent in a physical sense but must also involve active respect towards the public and the opposition, regardless of their repressive responses" (Hallam 2019, p. 5). Our understanding of who Jesus is and why he did what he did is projected into Jesus' internal states and motives. Jesus' vulnerability is as important as his reasons for accepting crucifixion. A sacrifice does not become sacred just because it is made with a pure heart. In the life of the apostles, sacredness is not just the moment of their calling, or their witness to the transfiguration, or their cowardice, or their witness to the resurrection, or their martyrdom: it is an ongoing process in which they are vulnerable in different and evolving ways.

However, the vulnerability of persons during civil disobedience is sacred, even if those persons do not explicitly connect with a prior model of vulnerability. In other words, civil disobedience is independent. Secondly, they are sacred, even if the individual's motives for undertaking civil disobedience are hidden or mixed: in other words, individuality is not the point of importance. The analysis we offer of these two types of sacredness is independence. XR's acts of civil disobedience are sacred not because they adhere to an ideology of sacredness, but because the vulnerability itself is sacred. The exposure that the participants have to each other and in the face of authorities is deep. The vulnerability begins with the experience of physical threat. Participants often get hurt either because of how they choose to be vulnerable (locking on to a stationary object, gluing themselves to something, being exposed to the weather) or the way that they are treated by authorities (pepper-sprayed, truncheoned, thrown, or moved aggressively). There is also intense psychic and emotional vulnerability. The strain before, during, and after the action (and arrest) carries over for weeks or months, placing immense tension on the person[26]. Civil disobedience also strains the imagination, because the experience of vulnerability here re-exposes you to the significance of climate change and the necessity of the act you are undertaking. In other words, no additional qualifications of the act are needed to contextualize its significance. If this sacredness resonates with other cycles of sacredness, it is because of the integrity, the self-containedness, of the vulnerable act.

Second, the individual's specific vulnerability or their motives neither curtail the meaningfulness of the act nor enhance it. The individual's vulnerabilities are shared with everyone around them. Individual vulnerability and motives are shaped by the vulnerability of the collective and the collective intention. In other words, no single individual or single ideology defines the vulnerability of civil disobedience and even if one individual or collective motive were present prior, motives and vulnerabilities are reshaped by the event. In this way, the self-containedness of the sacredness of the act of civil disobedience is supported and transformed by the collective way the actions are experienced.

---

25　(https://melbourneactivistlegalsupport.org/2019/12/06/report-the-policing-of-the-imarc-protests/.)
26　(https://rebellion.earth/act-now/resources/arrestee-welfare/ accessed on 16 May 2020).

Regenerative Culture (hereafter Regen) attempts to understand, moderate, and support this vulnerability.[27] During actions, XR Actions Support Training[28] describe how to support actions to ensure that individuals do not become too exposed for too long. However, the movement in Melbourne struggled to identify where this vulnerability would happen. In Melbourne, Regen was initially connected to long-term, interventional, or clinical ideas of well-being. However, when civil disobediences occur, Regen provides action support—a form of triage for crises within the movement itself. However, the Spring Rebellion of October 2019 was a week of civil disobedience. It took place in many cities around the world. In Australia, there were rebellions in every state capital. Many actions were small, although some had thousands of participants. The larger UK Actions involved prolonged occupations of public spaces in London. In Victoria, Regen provided support for these actions and helped to keep participants and onlookers safe. Wellbeing began with essential elements: water, food, health (first aid), well-being (check-ins, guided periods of silence), and love. Collective vulnerability can also be a collective burnout or trauma. The care that the XR community requires is complex because the context wherein the traumatic experiences are occurring are complex (Nugent et al. 2014). XR activists, like other activists, experience forms of trauma and burnout (Hall 2014)[29]. The sources of this are civil disobedience, and consistent engagement with the narrative of cataclysmic climate change. In Melbourne, our psychologists used trauma-informed practices to build resilience, especially narrative therapy (Lely et al. 2019). We used the "Tree of Life" exercise, which is a guided reflection on our past, present, and future that focuses on helping vulnerable persons connect with their community and process the trauma not as an event that has affected or happened to an individual, but trauma that is experienced collectively and shared by the community (Denborough 2008, pp. 71–98). XR is not Christian, nor religious: it is the vulnerability that lies at the heart of the movement that resonates the sense of the sacredness many people find in it.

### 4.3. Vulnerability within Community: Regen and S.O.S.: Regenerative Practice and Deliberative Democracy

This section discusses the vulnerability that arises in community. It notes how community is built both around the challenges of meeting climate change and mass extinction as activists, as well as via the experience of vulnerability through nonviolent communication and deliberative democracy. XR sees that any development of interiority emerges within the relationships between the truth of ecological breakdown, civil disobedience, and community. Crudely, civil disobedience says stop, while Regen helps people to maintain civil disobedience by proclaiming: let us change our relationships to each other. A community that accepts apocalyptic narratives and acts on them through civil disobedience is vulnerable in the different ways discussed previously. As discussed in the first section of the paper, civil disobedience can produce several effects. Healthy personal relationships and communication are essential for maintaining the activist community. Communal love and respect are both essential in themselves and an important antidote to burnout. For the community to stay healthy and resilient, groups need a culture that is regenerative. This section discusses Regen in XR and how it aims to build community. Using examples, we discuss how Regen evolved into a practice of being vulnerable together in community.

Regen supports people so that they can maintain civil disobedience, but it also has a vision for the transformation of cultural elements, which it identifies as dysfunctional and alienating. Regen's broader "mission", then, is to enshrine a culture of respect and love: firstly, within civil disobedience and, secondly, in the ways that XR members treat each other, resolve tensions, and the creative innovative ways to be in communion today. Activism is considered occasional rather than a way of life. Resistance to climate change and mass extinction can only cease when the world is no longer

---

[27] (https://www.youtube.com/watch?v=WCboH6kMuHs accessed on 16 May 2020).

[28] (https://rebellion.earth/act-now/resources/wellbeing/ accessed on 16 May 2020).

[29] See also: Post Traumatic Stress Disorder (PTSD) And Activists. *Activist Trauma Support*. https://www.activist-trauma.net/en/mental-health-matters/ptsd.html accessed on 16 May 2020. https://www.activist-trauma.net/ (This website is inactive).

threatened by climate change and mass extinction; therefore, the activism of XR members is often conceived of as more a way of life than something sporadic.

Below is a brief example of how Regen in XR began and some of the ways it evolved. Although it draws on multiple sources for its views on Regen, the movement tends to prefer a direct response (civil disobedience) to climate change. Regen focuses on activist wellbeing. When people are stressed, they are less likely to be able to sustain postures of love and respect in relation to law enforcement and members of the public. Regen also understands that the processes needed to support activist wellbeing are broader than simply administering first aid and facilitating conflict resolution. Regen can involve dealing with different types of trauma, wellbeing practices and nonviolent communication, conflict resilience, and empathy. Hence, Regen involves widening circles of support. After the "Spring Rebellion", conflict resolution, nonviolent communication, and "empathy circles" (Rutsch 2019) [30] became foundational to the movement's health. Activism is stressful, and activists are often committed people who have strong, often divergent opinions on how to do things. Since its inception, XR Regen groups around the world have become more focused on communication between persons. Practical matters—such as how we talk to each other, how we make decisions, and how responsibilities are allocated—have become more and more central to the meaning of a "regenerative culture".

The practices that underpin the resilience of the early Christian community are more defined, whereas XR's are still emerging. We note that XR and other responses to Covid-19, in which XR groups took an active role in supporting vulnerable and isolated persons in the UK, may still reshape how activist communities relate to the broader community.

"Active hope" describes the necessity of moving from theoretical models of hope to practical actions undertaken not only by individuals, but also being undertaken within the community (Macy and Johnstone 2012, p. 127). Love must be practical; it must be active. The emphasis of love is on the practice of building resilient communities. The short history of Regen in XR has involved moving from a general sense of how wellbeing and resilience play out in an activist setting and developed and implemented practical means for facilitating the wellbeing of the members of the movement. Regen is one of the places in the movement where this process is being worked out in practice (Macy and Johnstone 2012, p. 122).

S.O.S. is not the use of tools for supporting deliberative democracy, or a place to coordinate the energy of activists: it is an attempt to make a self-sufficient activist community. The gestures of food sharing, conflict resilience, nonviolent communication, and collective education are practical antidotes to issues that commonly cause breakdowns in communities. However, we described XR as an activist movement first before it was a spiritual movement. This is important, as the basis of its sense of sacredness is due to how the movement experiences and expresses vulnerability within the act of resisting an apocalyptic scenario. Vulnerability is expressed externally in civil disobedience, interdependently within community, and internally. Our contention is that XR needs to develop a sense of interiority and reflection, and we argue that the experience of vulnerability within these areas must also be linked with two moments: (1) the moment in which a person experiences vulnerability and its expression; (2) the moment when vulnerability is reflected upon in light of a sense of sacredness. In other words, (1) is to be watchful and (2) is to probe the significance of what is being watched. In this way, we are awaiting these events to show us where and how God is dwelling within the apparent messiness.

## 5. Conclusions

The essay examined some contemporary engagements with climate change that stress the immanence of the issue and argued for an urgent response. The essay then examined instances where the early Church engages with a context that the community understood in apocalyptic terms.

---

[30] https://www.facebook.com/groups/XREmpathy/ accessed on 18 May 2020.

We found that XR and early Christianity relate to dynamic periods of history by taking distinctive forms of response. First, Jesus rising brings with it implications so dramatic that non-believers come to believe and respond by transforming their behavior. Second, for XR, the reality of the climate and ecological crisis necessitates taking personal responsibility for the situation and acting to intervene. In the two instances of the early Christian community and XR, both operate out of an acknowledgement of the reality of circumstances and respond in practical ways. Both movements sometimes experience events of their times through an apocalyptic lens. In both movements, love is not a mere side note, but the way that the community responds to its times. Each movement provides what it feels is the best way to respond to crisis: a mass commitment to collective love and deep vulnerability as the only alternative to extinction.

As a reflection, the authors believe that love is not about functioning in the world with a false reality; love is a vulnerable encounter with reality. Emerging from this encounter, the individual does not fall into despair: rather, they feel that despite the darkness in the world and themselves, a new life is possible.

XR's relationship to its own community is that the demands of sacrifice for the sake of the community evolve arounds acts of civil disobedience. For a participant in XR actions, to engage in civil disobedience is to act as a witness against the paradigm of "business as usual", which XR equates with death. In place of the person of Jesus, instead of a savior, we are seeing the revelation that certain kinds of human activity risk the destruction of life on earth. We are vulnerable to this truth, and actively attempt to move in the direction of non-destruction.

Participation in civil disobedience helps you to rediscover the vulnerability that lies at the heart of the Christian experience and can be transformative. Terry Eagleton discusses the relationship between the Christian faith and acts of resistance, railing against the present state of the churches (Eagleton 2009). His reflections on sacrifice repeat the same causal order: faith and the sacrifice of Jesus is the model that affects all sacrifices. However, it can work the other way, too. For a Christian, being involved in the vulnerability of XR can transform how they experience their faith. Although the contexts are non-Christian and non-religious, *quidquid recipitur ad modum recipientis reciptur*—"whatever is received is received according to the capacity of the receiver".[31] Hence, the experience of vulnerability to apocalyptic climate change, civil disobedience, and to other members of the movement gives rise to an emergent sense of sacredness, which resonates with the Christian participant.

My sincere gratitude to Nadira Wallace, a legend of a rebel, based in the UK, for all her work, and for inspiring and guiding the paper.

**Funding:** This research received no external funding.

**Conflicts of Interest:** The author is a member of Extinction Rebellion.

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
