# Peer review of "Responses to Apocalypse: Early Christianity and Extinction Rebellion"

_religions, doi:10.3390/rel11080384_

Round 1
Reviewer 1 Report
There are not empirical data about the establishment and the activity of this mouvement, and scientific quotations are inserted into the text quite randomly.
Morever, the Extinction Rebellion can not be considered as a religion. So, I am not able to find a basic scientific foci in the text.
In this sense, arguing that XR is a religion can be novel hyphotesis, on which the author could work in another article.
Author Response
I made some changes to the initial sentences to clarify our claims re. religion and XR.
The other comments were on the general use of references. I would need the editor to make a fuller explanation before I would be able to address their concerns appropriately.
Many thanks
Cullan
Reviewer 2 Report
This paper represents a great improvement from the version I first read. There are possibly three or four places where I suspect a little more proof reading is needed, and I still have problems with the use of 'I' in the paper.
Author Response
I have made the changes requested (to the best of my capacity). With regard to the use of I, I made changes where I could find any use, but additional line references from the editor would assist me here.
Many thanks
Reviewer 3 Report
The section of the paper named "How did early Christians understand the apocalypse?" needs to be better referenced. I find this section superficial.
Also, the author should describe (at least briefly) the distinction between apocalypticism and eschatology in the Bible and early Christian writings. It is not true that we only find apocalyptic views in the New Testament.
It appears that a part of the sentence on line 341 is missing.
The first paragraph of the paper's conclusion on lines 515 - 518 present two disparate statements the relatedness of which is dubious. Though these statements in themselves may be defended, the author has not made it clear enough how they are related or what they find themselves in a juxtaposition.
Author Response
I added references to the sections recommended and made efforts to clarify and connect/ contrast apocalypse and eschatology. The distinction, unfortunately, is very fine, at times the terms have been used interchangeably in some secondary sources. I am happy for further comments if my changes have not been adequate. I made changes/ additions/ deletions to the sections missing sentences or being incomplete, I have also excluded some of the comments/ points mentioned in the conclusion that were not sufficiently discussed in the content of the paper.
Round 2
Reviewer 1 Report
With the new initial sentences, the article appear quite good. I suggest to add in the second paragraph of the introduction 6-8 lines regarding the genesis of the movement. It is important to indicate informations concerning the founder, where the movement is settled, and the real number of members.
As for the rest, it can be published.
Author Response
I used XR's standard description of the founding of the movement. I have found some references to numbers of members from Australia. I am unable to find accurate numbers of members, internationally, due to A) number of active country groups, B) XR security culture (numbers come from members details within restricted data bases/ websites). I have provided indicative numbers from Australia.
This manuscript is a resubmission of an earlier submission. The following is a list of the peer review reports and author responses from that submission.
Round 1
Reviewer 1 Report
I agree this is an important area for research and explication. I do not see much eveidenc in what I have read that you as a thinker, writer, indeed as a participant in the relevant events, have realy come to grips with what is required by your choice to submit your thoughts to a scholarly journal. You require a much deeper knowledge of the apostolic and sub-apostolic age and theology as well as a strong control of contemporarary theology to make this paper work. It is entirely up to you whether you wish to invest the effort.
Reviewer 2 Report
1) The article would like to compare the Extinction Rebellion movement (XR) with the Early Christianity: there are no references regarding the social scientific study of Early Christianity (from historical, philosophical, sociological, or anthropological perspectives). To begin an exploration of the literature, I suggest this handbook:
https://www.amazon.com/Handbook-Early-Christianity-Science-Approaches/dp/0759100152
I think this is one of the main issues of the article, because the literature is focused only on the XR movement.
2) In general, there are also no historical, philosophical, theological, or sociological references. The author needs them for analyze the two case studies, and discuss them within scientific debates. The literature of the article is almost exclusively composed of websites. This situation is not acceptable.
3) The author does not indicate discipline(s) and methodology adopted in the research. The author can not just "offer considered reflections", but he has to develop basic disciplinary perspectives.
4) The structure of paragraphs and sub-paragraphs, as well as the division of the article's text, do not follow a scientific approach. The whole article is composed only of sub-paragraphs...! The author must re-define the structure of the article.
5) The references and quotations in the article do not follow the Harvard style.
6) The reconstruction of XR's historical development is not rigorous, and the definition of its religious teachings is not systematic.
7) The author uses several concepts without explaining their meaning, such as Regenerative Culture, Active Love, Civil Disobedience, Empathy Circles, or Nonviolent Communication.
8) The article does not offer real empirical data on XR movement.